# WeatherFLUX: Universal Weather Translation with Diffusion Models

## Abstract

Reliable camera-based autonomous driving needs a large number of road scene images across diverse weather conditions, but public datasets mostly contain sunny scenes and have few images for snow, rain, fog, and night. Prior work has attempted to expand the range of weather conditions through image translation, but in complex road scenes the target weather is often rendered unreliably and semantic content is not preserved. To address these limitations, we present WeatherFLUX, a diffusion framework for universal weather translation that learns from limited weakly paired data. The framework supports bidirectional translation between sunny and snow, rain, fog, and night while preserving semantic content. WeatherFLUX adopts triptych prompting, an in-context setup that stacks a reference image, a source image, and a blank canvas into a single input, helping the model learn what to change and what to keep. This setup ideally works with paired data, but in practice we rely on weak pairs rather than strictly matched pairs, so the model reflects the target weather style well yet can introduce geometric differences between the source and the output. To mitigate this issue, WeatherFLUX introduces three techniques that reduce these differences and improve consistency. First, image alignment reduces geometric differences between the source and the target before training. Second, frequency-aware prompting forms a fused conditional embedding that reflects weather style from the reference and semantic cues from the source. Finally, we add a semantic preservation loss to encourage structural agreement between the output and the source that helps maintain boundaries and layout. These components yield photorealistic and scene consistent translations with strong preservation of semantic content. Extensive qualitative results demonstrate that our method is highly competitive.

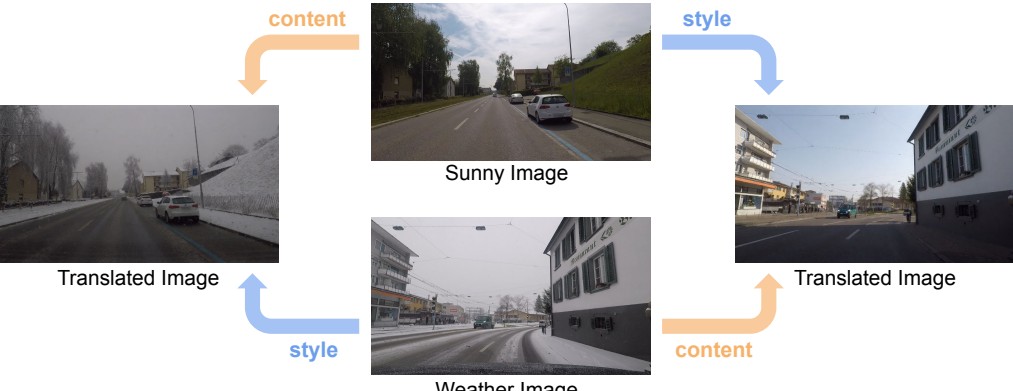

Figure 1: **WeatherFLUX** perform realistic bidirectional weather translation from sunny to snow, rain, fog, and night, and from each of these back to sunny, while preserving semantic content.

## 1 INTRODUCTION

Reliable camera-based autonomous driving needs large amounts of labeled road scene images across *diverse weather conditions*. Public benchmarks (Caesar et al., 2020; Sakaridis et al., 2021; Wilson et al., 2023; Marathe et al., 2023) offer limited coverage of these conditions and are biased toward sunny daytime scenes. A promising way to mitigate this scarcity is to translate labeled images captured in one weather condition into other conditions, thereby generating additional training data without manual annotation. Early GAN based approaches (Li et al., 2021; Song et al., 2023a) achieved partial success but still with significant limitations, such as unreliable weather transfer and imperfect preservation of semantic content.

Recent advances in diffusion models (Ho et al., 2020; Peebles & Xie, 2023) show strong capability for image synthesis and editing. These models achieve high-fidelity text-to-image generation (Saharia et al., 2022; Ruiz et al., 2023; Chen et al., 2023), image inpainting (Lugmayr et al., 2022; Song et al., 2023b; 2025), and image-to-image (I2I) translation. Given this progress, one might expect that diffusion-based methods could readily accomplish universal weather translation given their strength in style transfer, but this is not the case (Wang et al., 2023; Zhang et al., 2023b; Chung et al., 2024). In complex real-world images such as road scenes, style transfer methods often synthesize unrealistic weather effects or fail to preserve semantic content. Even recent studies on weather translation remain focused on narrow tasks such as removing specific weather effects or controlling lighting (Özdenizci & Legenstein, 2023; Cong et al., 2025; Lan et al., 2025).

In this paper we present WeatherFLUX, a framework for universal weather translation built on the FLUX backbone (Labs, 2024). Unlike prior work, WeatherFLUX performs realistic bidirectional translation from sunny to snow, rain, fog, and night, and from each of these conditions back to sunny, while preserving semantic content. To our knowledge this is the first work to cover all eight directions. For this task, we adopt triptych prompting from the polyptych paradigm (Shin et al., 2025; Song et al., 2025; Chen et al., 2025). A triptych stacks a reference image, a source image, and a blank canvas in one input and provides in-context guidance that text prompts or single image embeddings do not. This setting helps the model learn what to change and what to keep, taking weather from the reference and keeping structure from the source. Inspired by these in-context learning characteristics, we apply triptych prompting to weather translation, which we find well suited to the task.

However, applying this in-context setup to weather translation without modification does not yield satisfactory results. It ideally requires paired images of the same scene captured from exactly the same viewpoint with identical camera settings and differing only in weather. In real world driving scenarios, such pairs are infeasible to collect, so training relies on weakly paired data that show the same location but differ in camera pose and object placement, leading the model to learn not only the weather translation but also unwanted shifts in viewpoint and object placement.

To address this issue WeatherFLUX introduces three techniques that enforce local semantic consistency between source and output images and improve the fidelity of weather translation. First, we perform a coarse global alignment of the training images using correspondence matching to partially reduce geometric differences between the source and the target. Second, we form the conditional image embedding by mixing low frequency components from the reference for global weather style with high frequency components from the source for semantic content, which preserves semantic content while transferring style. Third, we add an additional training loss that enforces high frequency consistency between the output and the source during late denoising, preserving structure and reducing artifacts. Our contributions are summarized as follows:

- To our knowledge this is the first approach using a diffusion model that enables realistic universal weather translation through in-context learning with a small amount of data.

- To train with imperfectly paired data, we introduce three complementary techniques—image alignment, frequency-aware prompting, and semantic preservation loss—that together enhance translation quality and consistency.

## 2 RELATED WORK

**In-Context Learning for Vision.** In-context learning, first popularized in large language models (Brown et al., 2020), has been adapted to diffusion transformers (DiT) for image generation (Peebles & Xie, 2023), where DiT based text to image models exhibit in-context behavior. Recent work uses polyptych inputs that supply reference images as context and allow attention to flow across panels. Diptych Prompting (Shin et al., 2025) places a reference next to a masked target and strengthens cross panel attention for subject fidelity. Edit Transfer (Chen et al., 2025) composes an example and a query in four panels and fine tunes a small adapter to learn non rigid transformations from few examples. Insert Anything (Song et al., 2025) adopts triptych prompting, concatenating a reference, a source, and a masked canvas into a three panel input, which enables mask guided and text guided insertion while preserving identity and scene harmony. A training free variant (Kang et al., 2025) arranges copies of the reference in a grid and completes the missing panel, revealing in context behavior in a vanilla FLUX model (Labs, 2024). Together, these studies show that polyptych inputs provide both style cues and semantic guidance, enabling reference conditioned generation that respects style and content.

**Image-to-Image Translation.** Image-to-Image (I2I) translation maps images from a source domain to a target domain while preserving semantic content. Classic approaches include Pix2Pix (Isola et al., 2017) for paired supervision, CycleGAN (Zhu et al., 2017) for unpaired learning with cycle consistency, and Star-GAN2 (Choi et al., 2020) for multimodal transfer across domains. Diffusion-based methods further advance this line. InstructPix2Pix (Brooks et al., 2023) enables text driven edits, Plug-and-Play (Tumanyan et al., 2023) leverages diffusion features for controllable translation, and ControlNet (Zhang et al., 2023a) incorporates explicit structural guidance. In addition, StyleID (Chung et al., 2024) provides training free reference guided transfer by injecting reference features into self attention while keeping source queries, which preserves semantic layout without fine tuning.

**Weather Translation.** Weather translation aims to add or remove conditions such as rain fog snow and night while preserving scene semantics. Early studies used physics based simulation for controllable condition changes (Von Bernuth et al., 2019; Tremblay et al., 2021). Subsequent work synthesized adverse conditions without paired supervision to augment training and evaluation (Punnappurath et al., 2022). Generative translation then progressed from adversarial learning to transformer and diffusion architectures, improving controllability and structural consistency through attention control and explicit structure guidance (Song et al., 2023a; Zhu et al., 2024; Tumanyan et al., 2023; Zhang et al., 2023a). A recent diffusion approach further reduces transient artifacts during weather generation and strengthens consistency across frames and details (Zhu et al., 2025). In the inverse direction diffusion based restoration complements translation by improving visibility under night and haze (Özdenizci & Legenstein, 2023; Cong et al., 2025; Lan et al., 2025). Monocular nighttime simulation has also been explored to generate realistic night images for robust perception (Tzevelekakis et al., 2025). Despite these advances, achieving realistic weather translation while preserving semantic content remains challenging in complex road scenes.

## 3 PRELIMINARIES

**Triptych prompting** Following the in-context editing format of Insert Anything (Song et al., 2025) we adopt triptych prompting that concatenates a reference image, a source image, and an empty canvas to be synthesized. We assume $I_{\text{ref}}, I_{\text{src}} \in \mathbb{R}^{H \times W \times 3}$ are RGB images at a shared resolution, where $I_{\text{ref}}$ provides the target weather style and $I_{\text{src}}$ provides the scene content to be preserved. Let $\text{stack}_v(\cdot)$ denote vertical concatenation. The triptych input is

$$I_{\text{triptych}} = \text{stack}_v\big(I_{\text{ref}}, I_{\text{src}}, 0_{H \times W \times 3}\big) \in \mathbb{R}^{3H \times W \times 3}. \tag{1}$$

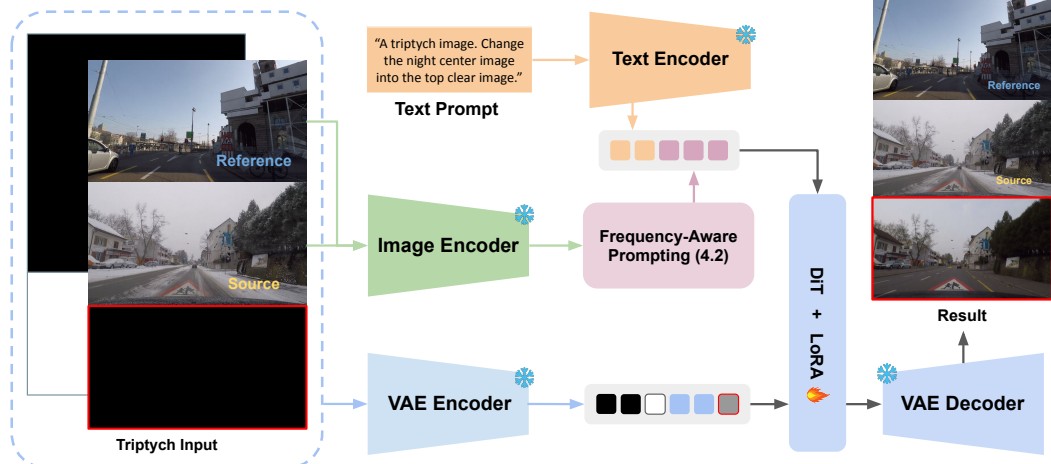

Figure 2: **Overview of WeatherFLUX.** Triptych input is encoded with a VAE encoder into a shared latent grid that places the three panels in one spatial frame, enabling cross panel attention in the DiT. Frequency-aware prompting produces fused features by combining reference style with source content. These features are concatenated with text prompt embeddings to form a single guidance signal, which conditions a DiT for in-context learning and enables realistic weather translation.

We also use a binary mask with the same vertical layout, where the top and middle panels are fixed and only the bottom panel is marked for synthesis. Let $M \in \{0, 1\}^{H \times W}$ and in our setting $M = 1_{H \times W}$. The full triptych mask is

$$M_{\text{triptych}} = \text{stack}_v(0_{H \times W}, \, 0_{H \times W}, \, M) \in \{0, 1\}^{3H \times W}. \tag{2}$$

The triptych provides explicit in-context visual guidance that text prompts or single image embeddings cannot offer. This reduces reliance on prompts or adapters and makes the approach well suited to complex road scene weather translation. The model takes weather cues from $I_{\text{ref}}$ such as lighting, scattering, and surface wetness, while $I_{\text{src}}$ anchors scene layout and object identities. However, under weakly paired supervision triptych prompting alone does not reliably preserve the semantic content of $I_{\text{src}}$, so we introduce additional techniques to stabilize semantics and maintain content fidelity.

## 4 WEATHERFLUX

WeatherFLUX adopts an in-context prompting scheme, formatting the input as a vertical triptych (**Section 3**). This explicit arrangement enables controlled style transfer from the reference image. However, this alone often results in insufficient preservation of local semantic details. To address this limitation, our framework integrates three additional components: **Section 4.1** image alignment to pre-align geometrical viewpoint differences between target and source, **Section 4.2** frequency-aware prompting that creates a fused token during prompt construction that carries semantic content from the source and weather style from the reference, and **Section 4.3** adds semantic preservation losses during training to enforce structural consistency. An simple overview of the framework is presented in Figure 2.

### 4.1 IMAGE ALIGNMENT

In real-world road scene datasets, perfectly paired images that capture the same viewpoint under different weather are difficult to obtain. Even with careful data collection along the same route, image pairs $(I_{\text{src}}, I_{\text{tgt}})$ often exhibit spatial offsets of several meters, creating a geometric misalignment.

Figure 3: **Overview of Frequency Aware Prompting.** Given encoded reference and source feature grids, we select low frequency components from the reference and high frequency components from the source in the Fourier domain and fuse them with IFFT to obtain a single set of fused feature grids, which reduces token count and encourages separation of weather appearance and scene content.

To reduce this geometric domain shift, we pre-align $I_{\text{tgt}}$ and $I_{\text{src}}$ using correspondence matching. For each pair, we use MASt3R (Leroy et al., 2024) to extract reliable matching points. We identify which image has more dense matching points, which usually corresponds to the image captured from a further spatial position. We then estimate a similarity transform matrix $A^*$ that best aligns this image to the one captured from a closer position. The transform is computed by minimizing the squared distance between corresponding points:

$$A^* = \arg\min_A \sum_i \left\| A\, P_i^{\text{tgt}} - P_i^{\text{src}} \right\|_2^2. \tag{3}$$

The image with the denser matching points is warped and cropped to maximize its structural overlap with its pair. After applying this transformation to all image pairs in the dataset, we perform an additional filtering step to ensure high structural quality. We automatically compute the Structural Similarity Index Measure (SSIM) score for each aligned pair. Only pairs with an SSIM score exceeding a predefined threshold are retained for the final training set. This two-stage process of geometric alignment followed by SSIM-based filtering allows us to curate a small but high-quality dataset of image pairs with minimal structural discrepancies, which is crucial for effective model training. For clarity, sample aligned image pairs are shown in Appendix A.3.

## 4.2 FREQUENCY-AWARE PROMPTING

Using full encoder features from the reference and the source as prompts increases the number of tokens and often causes the model to confound style cues with structural details, which weakens weather transfer and content preservation. Instead, we construct a single fused image prompt that combines weather style from the reference with content cues from the source. This compact prompt reduces the number of tokens and mitigates confounding between style and content. The overall procedure is shown in Figure 3.

To separate weather style from semantic content, we work in the frequency domain. Low frequencies tend to capture global appearance such as color and illumination. On the other hand, high frequencies tend to capture edges, boundaries, and fine layout. We extract low frequency components from the reference and high frequency components from the source and obtain fused features that serve as the image prompt.

Let $\mathcal{E}_{\text{img}}$ be the frozen image encoder of the backbone, which maps an RGB image of size $H \times W \times 3$ to a grid of $h \times w \times C$ visual tokens.

$$e_{\text{src}} = \mathcal{E}_{\text{img}}(I_{\text{src}}) \in \mathbb{R}^{h \times w \times C}, \qquad e_{\text{ref}} = \mathcal{E}_{\text{img}}(I_{\text{ref}}) \in \mathbb{R}^{h \times w \times C}. \tag{4}$$

Figure 4: **Overview of Semantic Preservation Loss.** We obtain $\hat{z}_0$ by one-step denoising from the noisy latent $z_t$ and decode it to $I_{\text{pred}}$. A Fourier high-pass filter $f_H$ extracts the high-frequency components of $I_{\text{pred}}$ and the source $I_{\text{src}}$. The semantic preservation loss $\mathcal{L}_{SP}$ minimizes the mean squared error between these components and is optimized jointly with the flow-matching loss $\mathcal{L}_{FM}$ at low-noise timesteps $0 \leq t < 200$, encouraging the output to preserve edges, object boundaries, and scene layout while modifying only the weather appearance.

Here $I_{\text{src}}$ and $I_{\text{ref}}$ denote the source and reference images, and $e_{\text{src}}$ and $e_{\text{ref}}$ denote their encoder feature grids, which we refer to as the source features and the reference features.

We use the 2D discrete Fast Fourier Transform FFT and its inverse IFFT applied per channel over the $h \times w$ spatial grid.

$$F_{\text{src}} = \text{FFT}(e_{\text{src}}), \qquad F_{\text{ref}} = \text{FFT}(e_{\text{ref}}). \tag{5}$$

We build a circular low pass mask $M_L \in \{0,1\}^{h \times w}$ centered at zero frequency with cutoff radius $r$, and set $M_H = 1 - M_L$. A larger $r$ passes more low frequencies from the reference, while a smaller $r$ preserves more high frequencies from the source.

$$F_L(e_{\text{ref}}) = F_{\text{ref}} \odot M_L, \qquad F_H(e_{\text{src}}) = F_{\text{src}} \odot M_H. \tag{6}$$

Here $\odot$ denotes elementwise multiplication. We compute fused features in the frequency domain and apply IFFT to obtain the fused feature grid

$$e^* = \text{IFFT}\big(F_L(e_{\text{ref}}) + F_H(e_{\text{src}})\big). \tag{7}$$

Finally, we concatenate $e^*$ with the text embeddings and use the fused features as the single image conditioning for WeatherFLUX. This compact conditioning reduces ambiguity and preserves the semantics of the source while transferring weather style from the reference.

### 4.3 SEMANTIC PRESERVATION LOSS

To better preserve the scene layout and boundaries of the source image $I_{\text{src}}$, we introduce a semantic preservation loss that is applied at low noise timesteps $0 < t \leq 200$, where the reverse process recovers high frequency structure rather than global style (Qian et al., 2024). We optimize the flow matching objective $\mathcal{L}_{FM}$ (Lipman et al., 2022) together with this term.

Let $z_t$ denote the input latent after adding noise at step $t$ and let $\hat{\epsilon}_t$ be the DiT prediction under our conditioning, where $\hat{\epsilon}_t$ represents the predicted noise. Since WeatherFLUX is trained with flow matching, we

estimate $\hat{z}_0$ by a one step backward Euler update from $z_t$ using $\hat{\epsilon}_t$. We work on a time axis $t \in (0, 1000]$ and use normalized time $s = t/T$ with $T = 1000$

$$\hat{z}_0 \;=\; z_t \;-\; s\,\hat{\epsilon}_t \;=\; z_t \;-\; \frac{t}{T}\,\hat{\epsilon}_t. \tag{8}$$

We decode $\hat{z}_0$ with the VAE decoder to obtain $I_{\text{pred}}$. We then extract high frequency components from $I_{\text{pred}}$ and from the source image $I_{\text{src}}$ using a high frequency filter $f_H$ implemented in the Fourier domain and minimize

$$\mathcal{L}_{SP} \;=\; \big\| f_H(I_{\text{pred}}) - f_H(I_{\text{src}}) \big\|_2. \tag{9}$$

Here $f_H$ is a high pass filter that suppresses most low frequency energy and retains high frequency content. In practice we use a ring shaped mask $R_H$ in the frequency plane and define

$$f_H(X) \;=\; \text{IFFT}\big(\text{FFT}(X) \odot R_H\big), \quad R_H(u,v) = \begin{cases} 0.1 & \text{if } \sqrt{u^2 + v^2} < r, \\ 1 & \text{otherwise.} \end{cases} \tag{10}$$

The total objective is

$$\mathcal{L} = \begin{cases} \mathcal{L}_{FM} + \mathcal{L}_{SP} & \text{if } 0 < t \le 200, \\ \mathcal{L}_{FM} & \text{if } 200 < t \le 1000. \end{cases} \tag{11}$$

This objective encourages modification of weather appearance while preserving geometric structure and object boundaries, which improves content consistency.

## 5 EXPERIMENTS

### 5.1 EXPERIMENTAL SETUP.

**Implementation Details.** WeatherFLUX builds upon FLUX.1-Fill [dev], an inpainting model based on the DiT architecture. We fine tune WeatherFLUX with LoRA on the ACDC training set after applying the alignment preprocessing of Section 4.1. For each weather condition the model is trained for 5000 iterations. Other baselines are trained on the full ACDC train data under the authors default settings without preprocessing. Full implementation details are provided in Appendix A.1.

**Baselines.** We compare WeatherFLUX with four methods. InstructPix2Pix (Brooks et al., 2023) enables text-driven image-to-image editing, Insert Anything (Song et al., 2025) performs triptych inpainting, StyleID* (Chung et al., 2024) provides training-free, reference-guided transfer, and FLUX.1 Kontext (Labs et al., 2025) serves as a recent diffusion-based image-to-image baseline. An asterisk (*) denotes training-free methods throughout the paper and figure.

**Test Datasets.** We evaluate WeatherFLUX on ACDC test set (Sakaridis et al., 2021), which provides 500 paired examples per weather condition. Additional qualitative results on the Cityscapes validation set (Cordts et al., 2016), are reported in Appendix A.5.

**Metrics.** We report FID using the Clean FID (Parmar et al., 2022) which standardizes preprocessing and yields more reliable scores. FID measures the Fréchet distance between Gaussian statistics of Inception features for generated and reference images. In addition we use ArtFID (Wright & Ommer, 2022) defined as $(1 + \text{LPIPS}) \times (1 + \text{FID})$ which balances content preservation and style fidelity and aligns well with human perceptual judgments. LPIPS (Zhang et al., 2018) measures content fidelity between the output image and the corresponding source image by computing a perceptual distance in deep feature space with a pretrained network. Unless stated otherwise the FID term in ArtFID is computed with the same Clean FID setup.

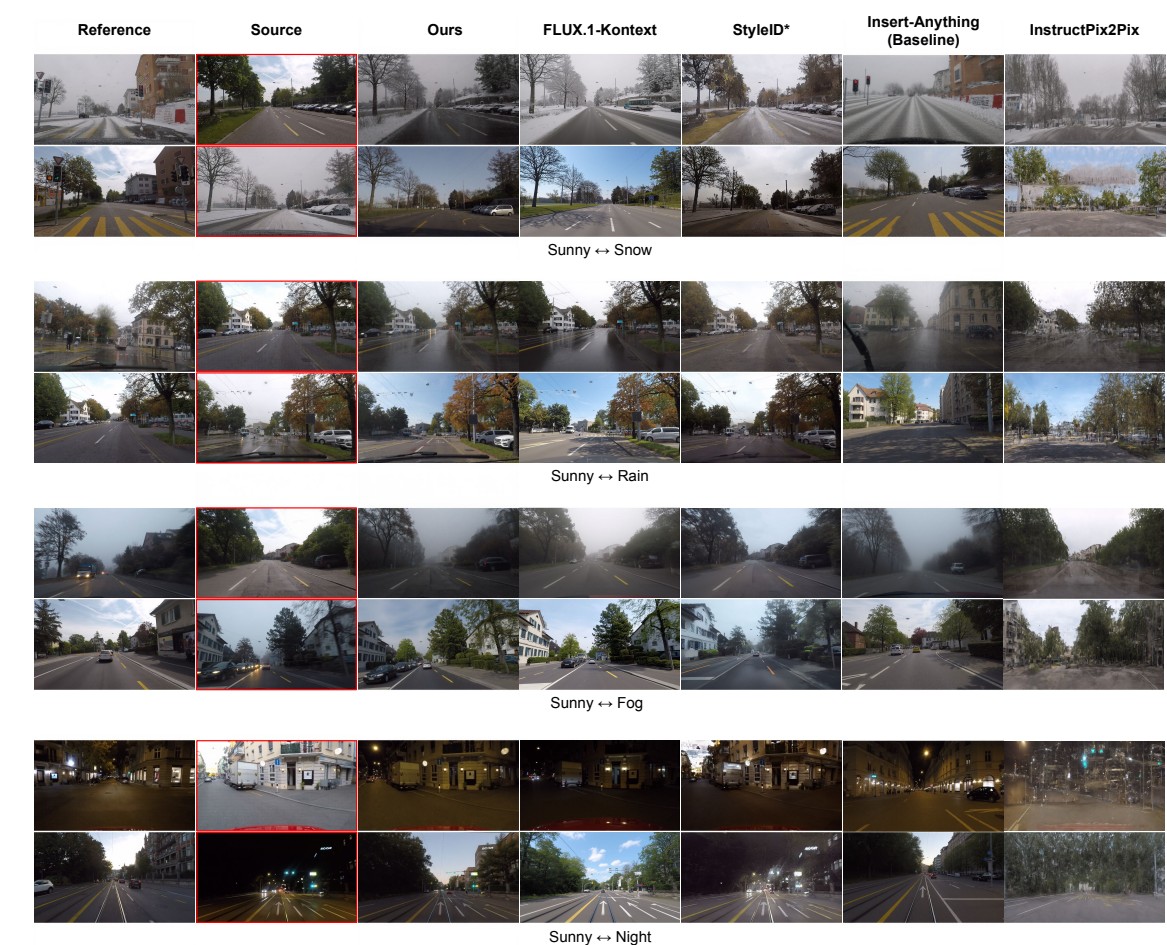

Figure 5: **Qualitative comparison on ACDC dataset.** Each top to bottom pair rows mean night ↔ day, snow ↔ sunny, rain ↔ sunny, fog ↔ sunny. WeatherFLUX achieves realistic weather translation while preserving source semantic content and scene layout.

## 5.2 QUALITATIVE RESULTS

Our qualitative evaluation comprises two parts. First, Figure 5 shows bidirectional weather translations between sunny and each of snow, rain, fog, and night. Second, to assess preservation of source semantics, Figure 6 compares the segmentation maps of the source image and its translation, computed with a pretrained SegFormer (Xie et al., 2021). For the segmentation analysis, we show a representativ sunny→rain example. Additional qualitative results are provided in Appendix A.4.

Across all conditions, WeatherFLUX faithfully transfers the target weather while preserving the source semantic layout and object boundaries, yielding photorealistic textures and strong semantic consistency. StyleID preserves source semantics and layout very well, but produces collage-like artifacts in the rendered weather. FLUX.1-Kontext and Insert Anything reflect the target weather style well, yet they fall short in preserving source semantics and layout, which reduces structural fidelity.

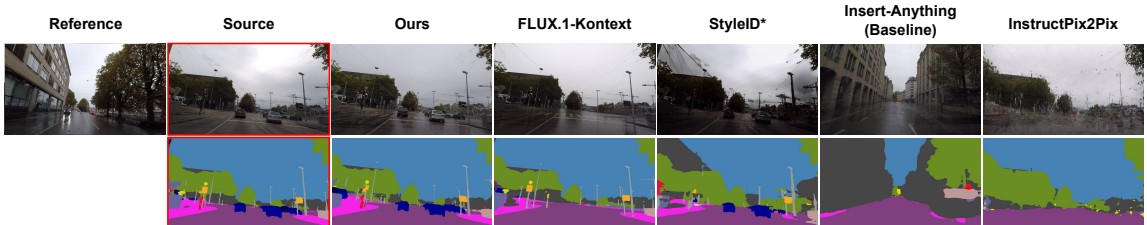

Figure 6: **Segmentation consistency comparison.** We present a single representative sunny→rain example. Segmentation maps are predicted with a pretrained SegFormer Xie et al. (2021). WeatherFLUX preserves the source semantic layout with high fidelity, avoiding object removal and false-positive mask creation.

## 5.3 QUANTITATIVE RESULTS

As shown in Tables 1 and 2, WeatherFLUX achieves the best ArtFID in every translation direction and the best FID in seven of eight directions. Performance on snow to sunny ranks second in FID. These results indicate faithful transfer of the target weather style with strong preservation of semantic content and scene layout. The effectiveness of the techniques introduced in Sections 4.2 and 4.3 is examined in Appendix A.2.

| Method | sunny → snow | | sunny → rain | | sunny → fog | | sunny → night | |
|---|---|---|---|---|---|---|---|---|
| | FID↓ | ArtFID↓ | FID↓ | ArtFID↓ | FID↓ | ArtFID↓ | FID↓ | ArtFID↓ |
| Inspix2pix(Brooks et al. (2023)) | 24.54 | 42.72 | 38.57 | 59.96 | 31.69 | 46.84 | 17.52 | 31.17 |
| Insert-Anything(Song et al. (2025)) | 20.28 | 32.55 | 14.51 | 25.91 | 6.18 | 11.70 | 11.27 | 21.55 |
| StyleID*(Chung et al. (2024)) | 18.30 | 25.08 | 10.67 | 14.39 | 15.58 | 21.20 | 19.58 | 29.03 |
| FLUX.1-Kontext(Labs et al. (2025)) | 6.55 | 11.68 | 7.95 | 12.30 | 5.09 | 8.74 | 9.20 | 16.98 |
| Ours | **5.92** | **9.78** | **7.62** | **11.35** | **3.28** | **6.13** | **6.08** | **11.38** |

Table 1: **Quantitative comparison on ACDC dataset.** (sunny → snow/rain/fog/night)

| Method | snow → sunny | | rain → sunny | | fog → sunny | | night → sunny | |
|---|---|---|---|---|---|---|---|---|
| | FID↓ | ArtFID↓ | FID↓ | ArtFID↓ | FID↓ | ArtFID↓ | FID↓ | ArtFID↓ |
| Inspix2pix(Brooks et al. (2023)) | 49.03 | 79.61 | 44.12 | 68.62 | 34.89 | 55.16 | 33.63 | 59.19 |
| Insert-Anything(Song et al. (2025)) | **7.65** | 14.76 | 13.66 | 25.19 | 4.72 | 9.63 | 11.50 | 21.89 |
| StyleID*(Chung et al. (2024)) | 17.03 | 22.87 | 11.50 | 15.25 | 18.56 | 23.93 | 17.40 | 26.38 |
| FLUX.1-Kontext(Labs et al. (2025)) | 8.95 | 15.35 | 8.67 | 13.73 | 10.75 | 17.85 | 10.16 | 18.41 |
| Ours | 8.53 | **12.92** | **7.02** | **10.78** | **4.19** | **7.17** | **5.73** | **9.63** |

Table 2: **Quantitative comparison on ACDC dataset.** (snow/rain/fog/night → sunny)

## 6 CONCLUSION

In this paper, we proposed WeatherFLUX, a diffusion-based framework for realistic and semantically consistent universal weather translation. By integrating image alignment, frequency-aware prompting, and semantic preservation loss, our method delivers photorealistic translations with strong structural fidelity. Experiments validate its effectiveness and show clear advantages over existing approaches. Beyond robust data augmentation for autonomous driving, WeatherFLUX provides a general framework that can be extended to other domains where preserving semantic content under diverse visual conditions is essential.

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

# A APPENDIX

## A.1 IMPLEMENTATION DETAILS

**Hardware and environment.** All experiments were run on a single NVIDIA A5000 GPU with 24 GB memory.

**Backbone and adaptation.** Our method builds upon FLUX.1 Fill [dev], an inpainting model with a DiT backbone. We apply LoRA with rank 16 to the transformer modules only.

**Quantization.** To operate under the 24 GB constraint, we quantize all modules of FLUX.1 Fill including the transformer, the image encoder, and the decoder to `qint8` using the `optimum-quanto` library.

**Training setup.** We use a batch size of 1 and process all images at a resolution of $768 \times 768$. Optimization uses the Prodigy optimizer (Mishchenko & Defazio, 2023) with safeguard warmup and bias correction enabled and with weight decay 0.05. Our model is trained on the ACDC train split after alignment as described in Section 4.1. For each weather condition the model is trained for 100 epochs. Other models are trained on the full ACDC train split under their default settings unless stated otherwise.

**Objective and sampling.** The training loss follows the flow matching objective (Lipman et al., 2022). During sampling we perform denoising for 50 iterations.

## A.2 EFFECTIVENESS OF FREQUENCY-AWARE PROMPTING AND SEMANTIC PRESERVATION LOSS

We evaluate the contribution of frequency-aware prompting 4.2 and semantic preservation loss 4.3 beyond image alignment. Except for the baseline, image alignment is used as a fixed preprocessing step across all variants. We train four variants on ACDC with bidirectional translations between sunny and fog. The baseline Song et al. (2025) excludes both frequency-aware prompting and semantic preservation loss and does not apply image alignment. The first ablation removes the semantic preservation loss while keeping frequency-aware prompting, and the second removes frequency-aware prompting while keeping the semantic preservation loss. Our full model incorporates both components. Evaluation with FID and LPIPS demonstrates the individual benefits of each component and highlights the complementary gains achieved when they are combined.

| Method | fog → sunny | | sunny → fog | |
|---|---|---|---|---|
| | FID↓ | LPIPS↓ | FID↓ | LPIPS↓ |
| Baseline (Song et al. (2025)) | 4.72 | 0.68 | 6.18 | 0.63 |
| Frequency-aware prompting | 4.04 (**-0.68**) | 0.38 (**-0.30**) | 3.05 (**-3.13**) | 0.46 (-0.17) |
| Semantic preservation loss | 4.36 (-0.36) | 0.41 (-0.27) | 3.36 (-2.82) | **0.41 (-0.22)** |
| Ours | 4.19 (-0.53) | 0.38 (**-0.30**) | 3.28 (-2.90) | 0.43 (-0.20) |

Table 3: Ablation study on the ACDC fog test set. Values in parentheses show change from the baseline where lower is better

### A.3 SUPPLEMENTARY MATERIAL ON IMAGE ALIGNMENT

We show two complementary views to support section 4.1. Figure 7 presents image aligned pairs with matching points. Matching points are obtained with MASt3R (Leroy et al., 2024). Figure 8 presents the same pairs without matching points.

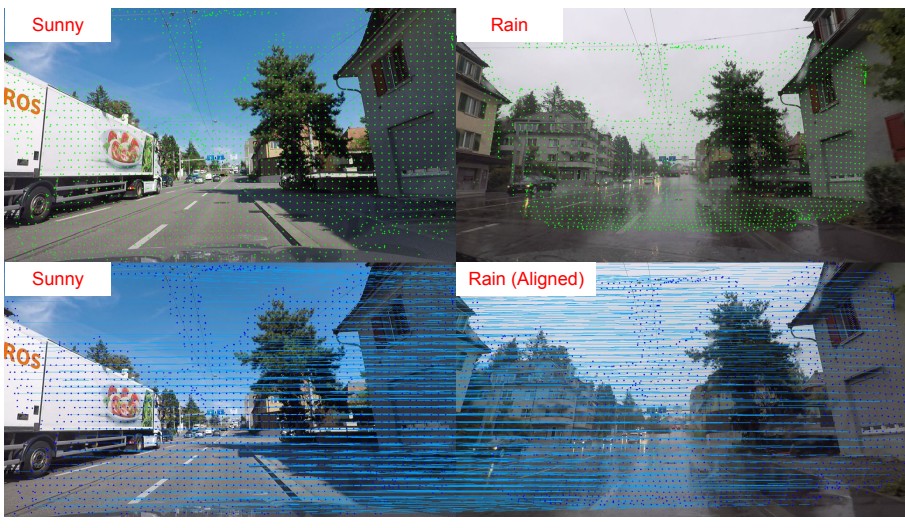

● Matching Point (Before Alignment)     ● Matching Point (After Alignment)

Figure 7: Image aligned pairs with matching point.

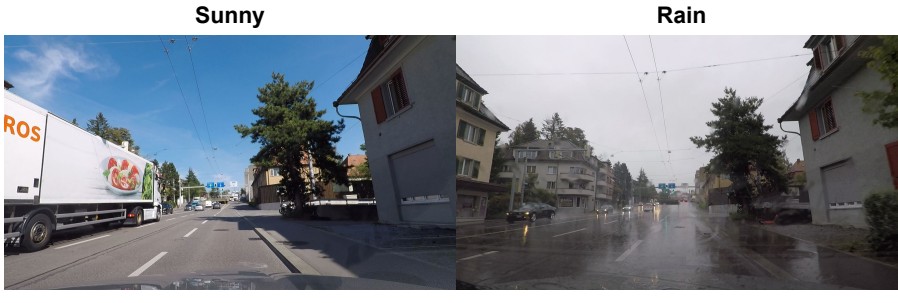

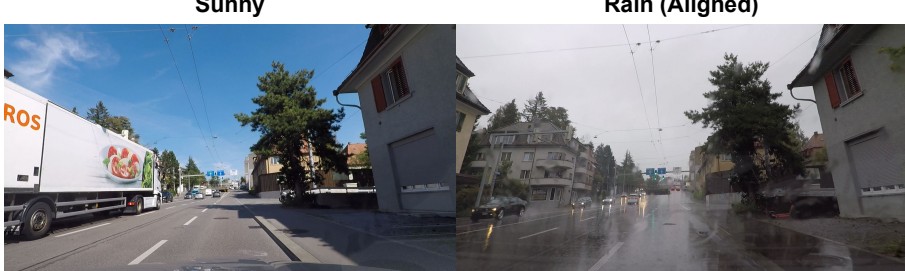

Figure 8: Image Aligned Pairs.

### A.4 SUPPLEMENTARY QUALITATIVE RESULTS

We provide additional qualitative results on the ACDC test set that are not included in Figure 5 and 6. The examples cover all eight translation directions between sunny and snow, rain, fog, and night.

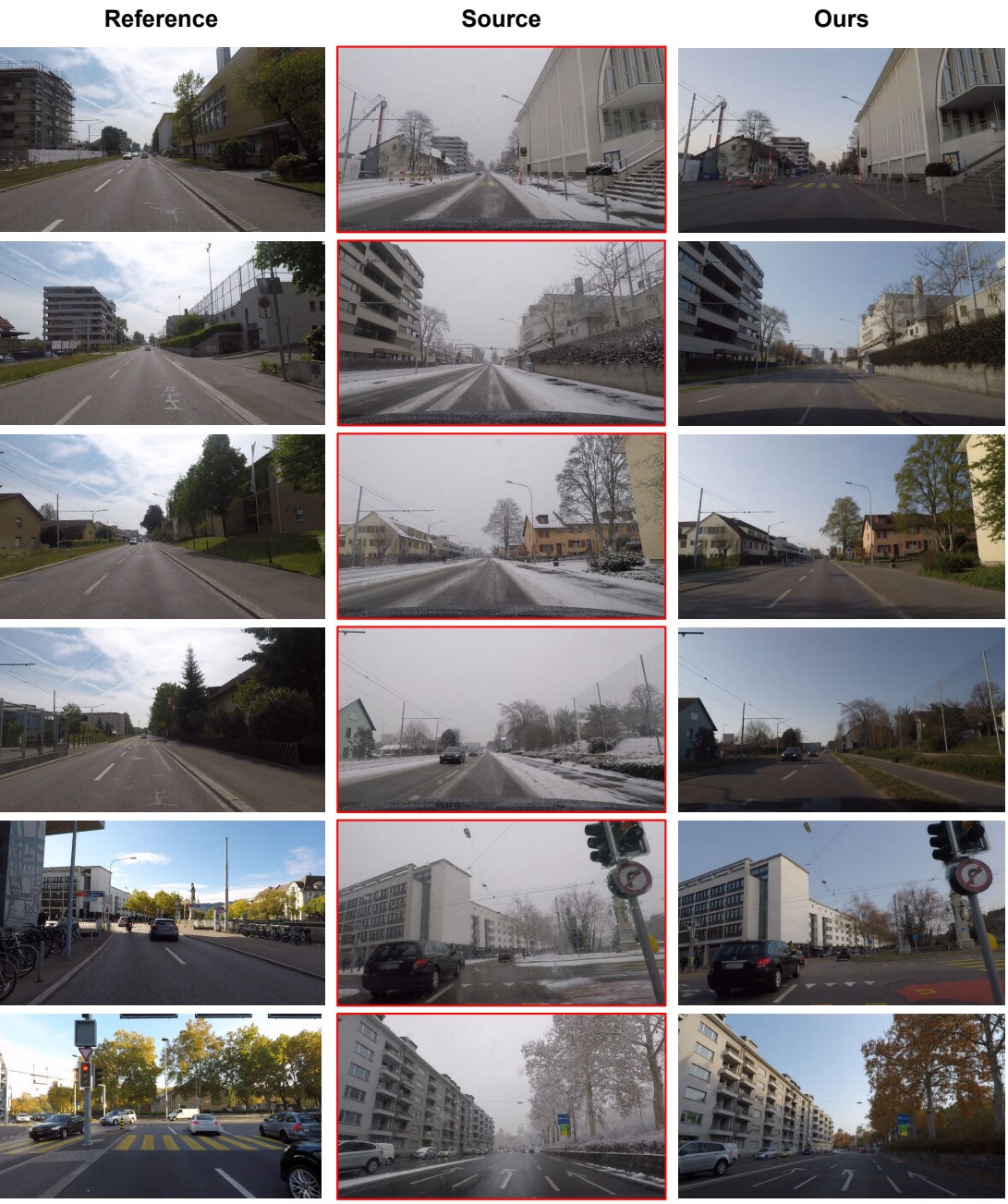

Figure 9: Snow to sunny weather translation on ACDC test set.

**Reference**    **Source**    **Ours**

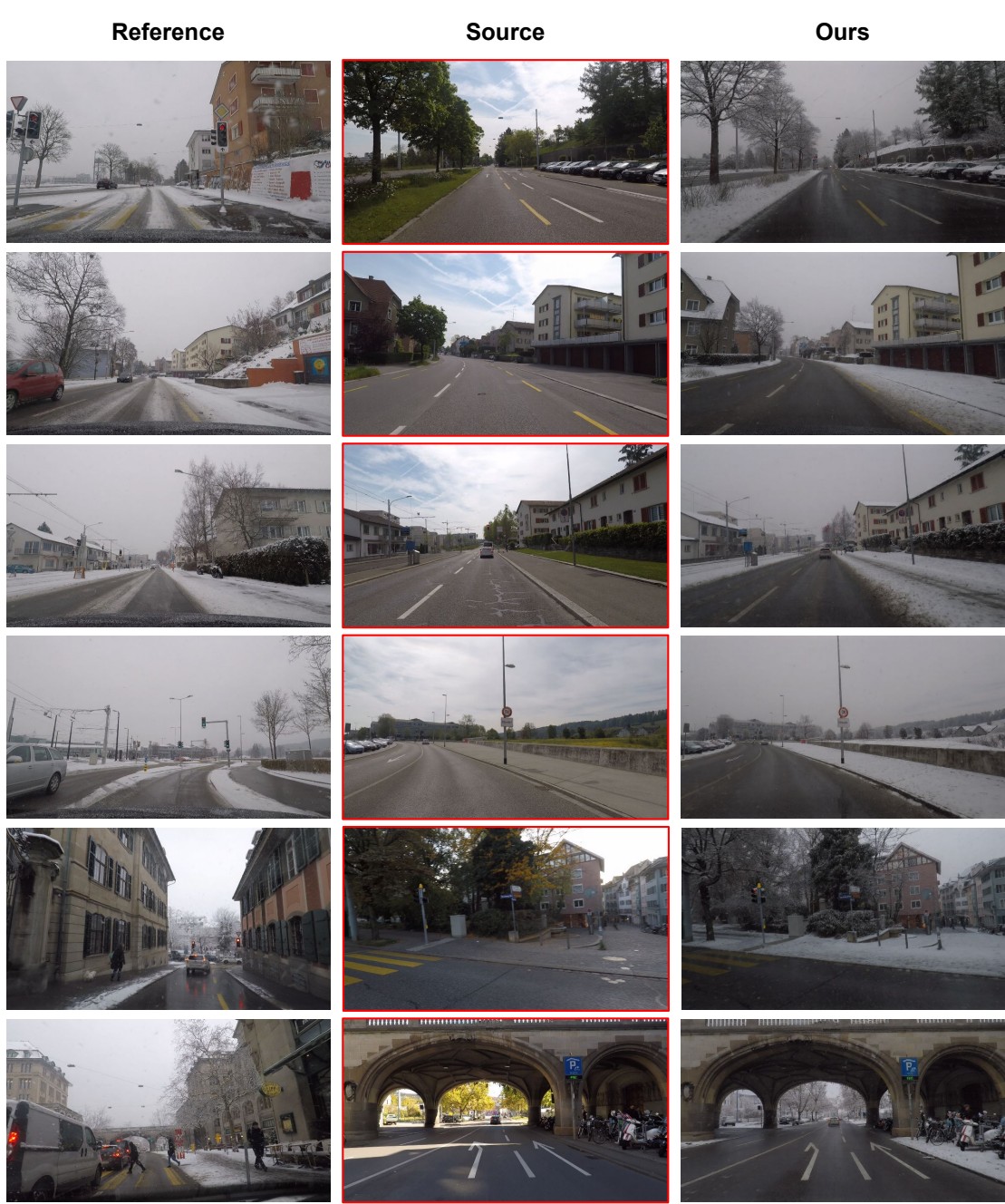

Figure 10: Sunny to snow weather translation on ACDC test set.

**Reference** **Source** **Ours**

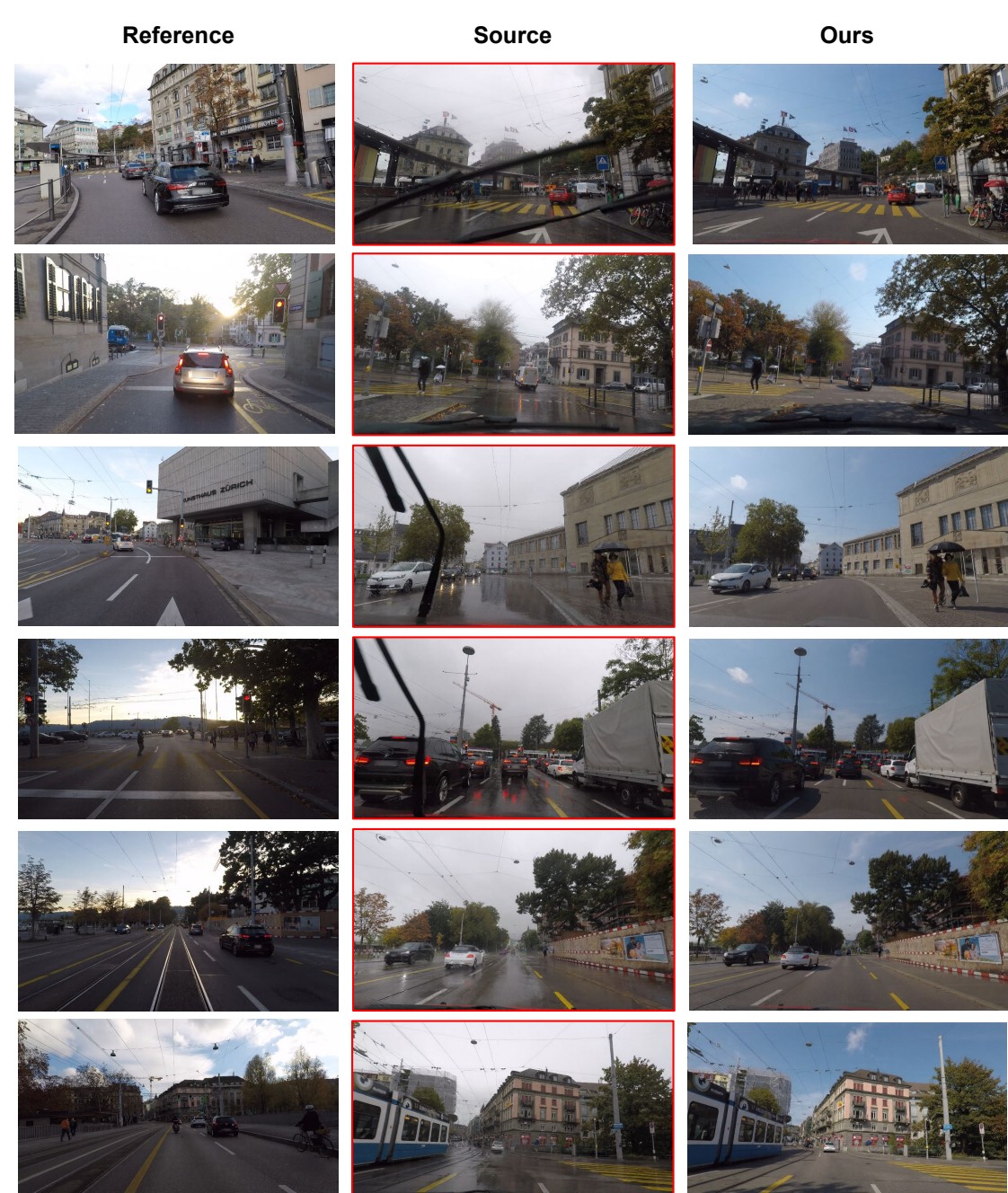

Figure 11: Rain to sunny weather translation on ACDC test set.

**Reference**  **Source**  **Ours**

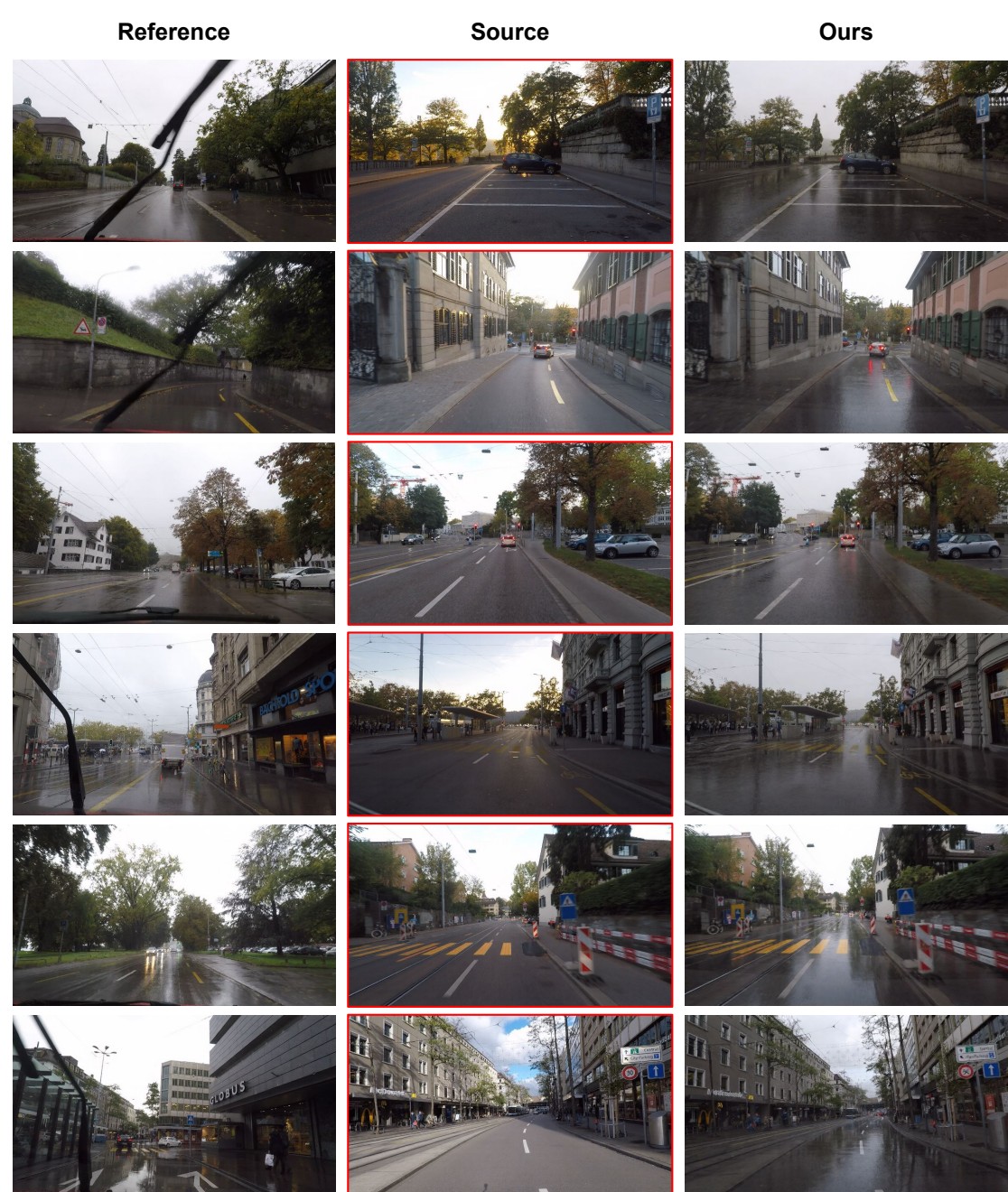

Figure 12: Sunny to rain weather translation on ACDC test set.

Reference          Source          Ours

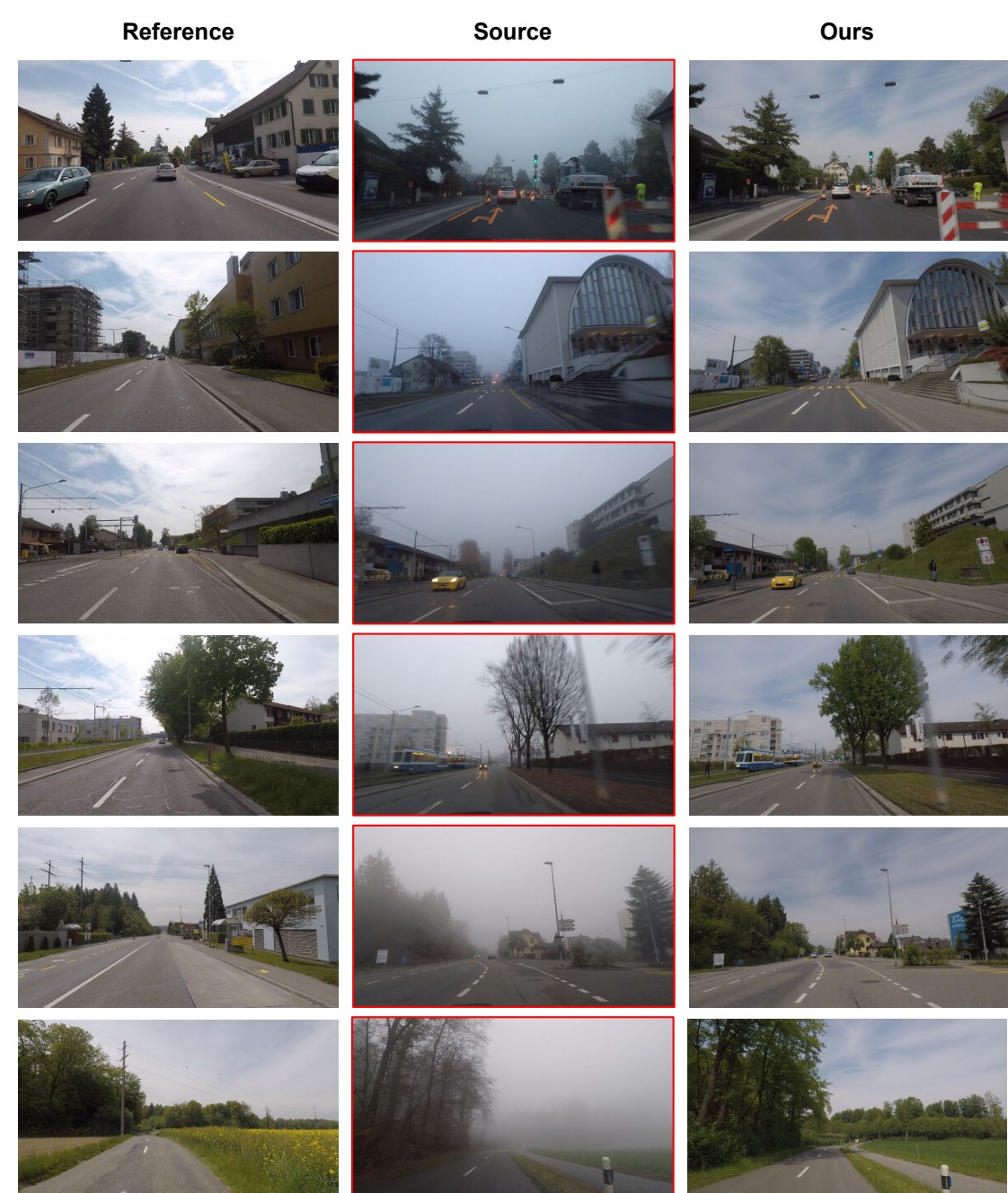

Figure 13: Fog to sunny weather translation on ACDC test set.

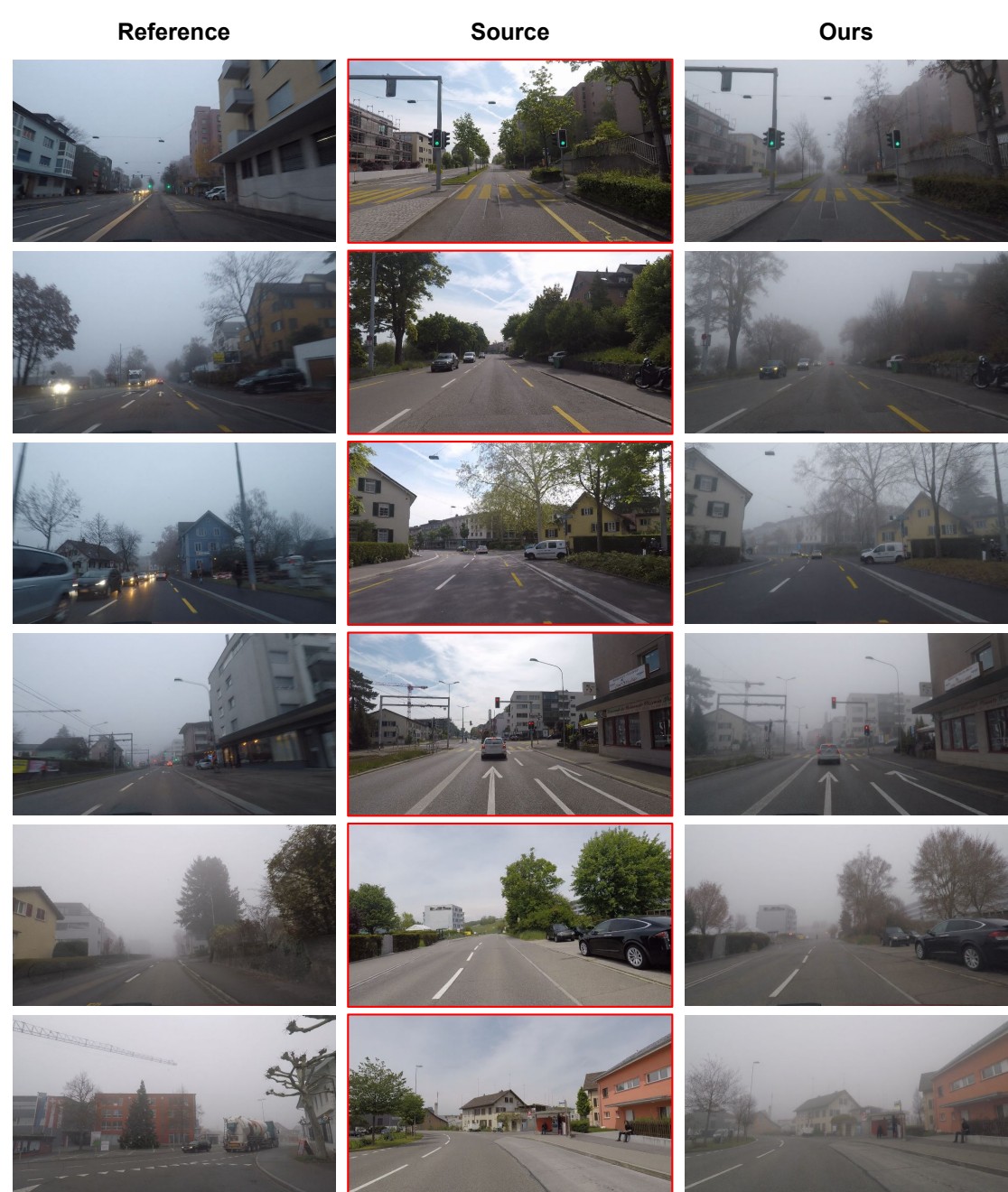

Figure 14: Sunny to fog weather translation on ACDC test set.

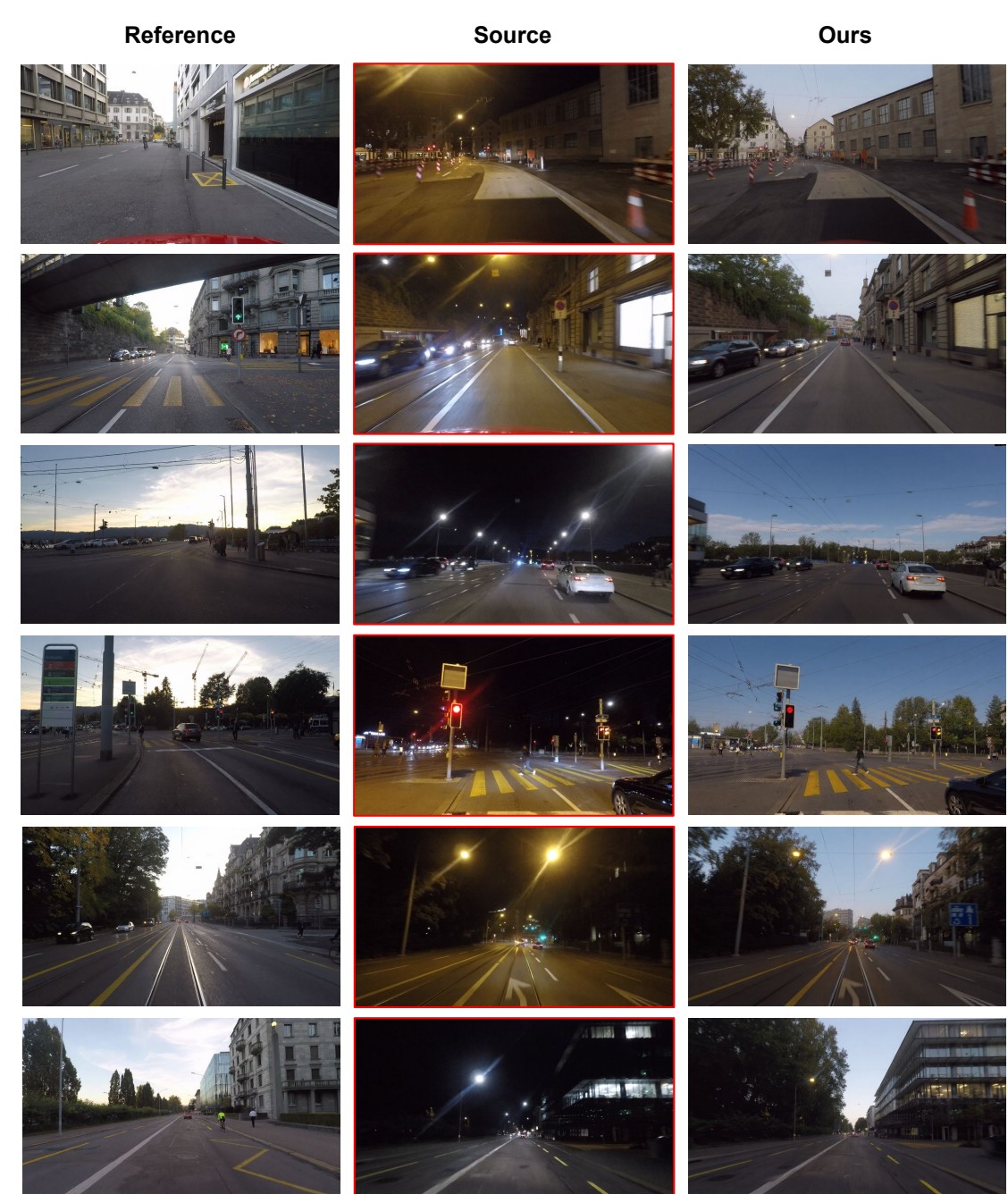

Figure 15: Night to sunny weather translation on ACDC test set.

**Reference**  **Source**  **Ours**

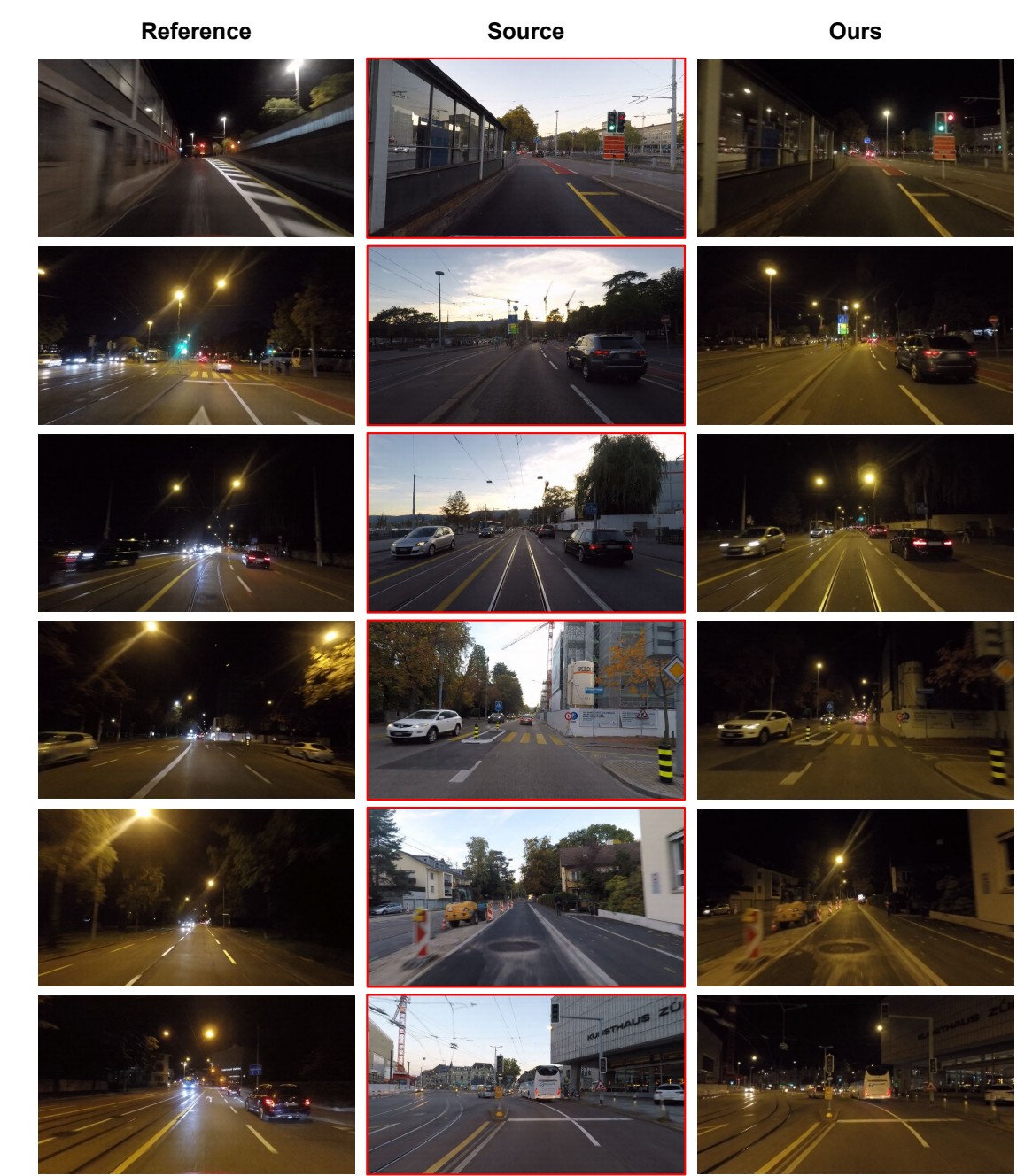

Figure 16: Sunny to night weather translation on ACDC test set.

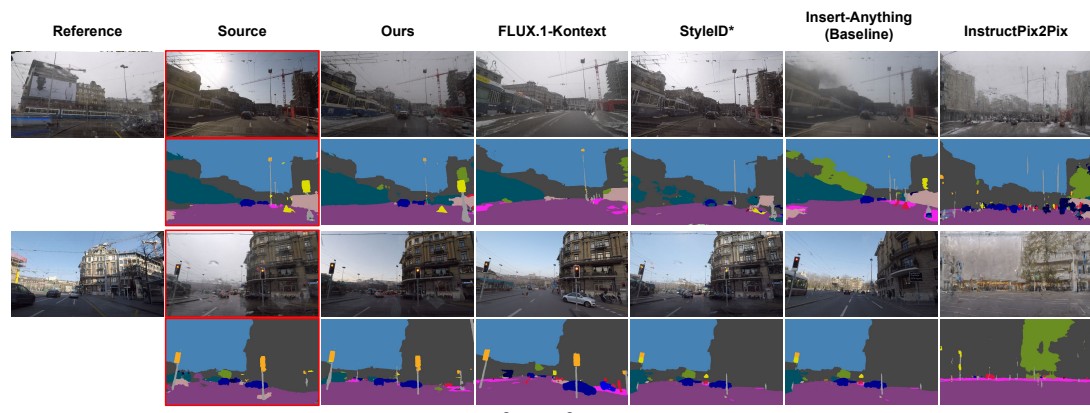

Figure 17: Segmentation consistency comparison. (sunny↔snow)

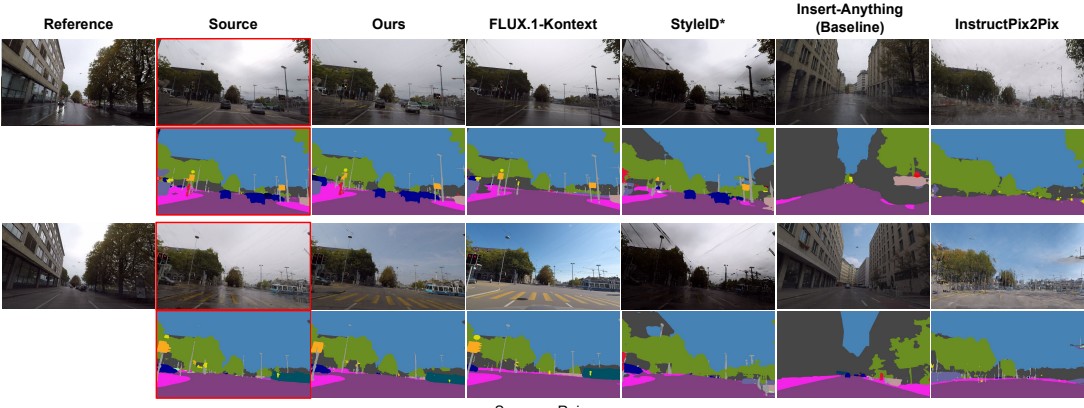

Figure 18: Segmentation consistency comparison. (sunny↔rain)

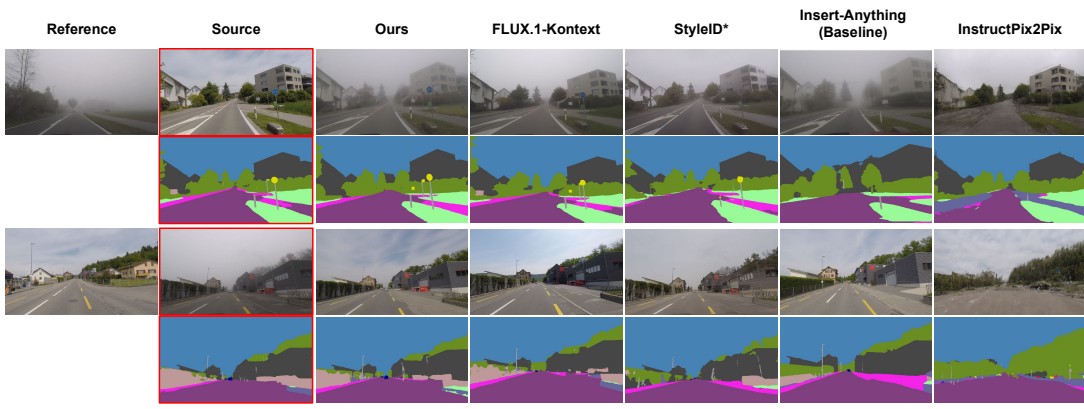

Figure 19: Segmentation consistency comparison. (sunny↔fog)

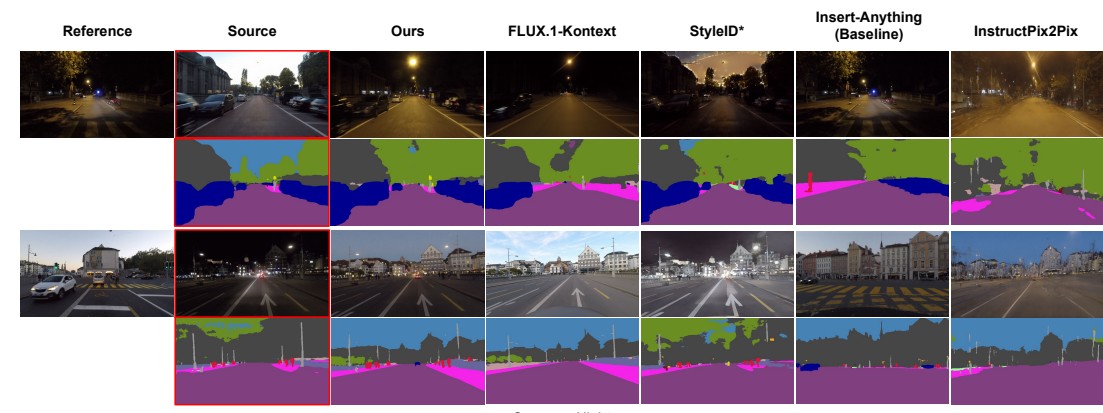

Figure 20: Segmentation consistency comparison. (sunny↔night)

## A.5 ADDITIONAL QUALITATIVE EXPERIMENTS

This appendix presents additional qualitative results on Cityscapes Cordts et al. (2016). All models except StyleID were trained only on ACDC, and no Cityscapes images were used during training. We inference on the Cityscapes validation set, and Figure 21, 22, 23, 24, 25 show that WeatherFLUX works well on Cityscapes even though it was not used for training.

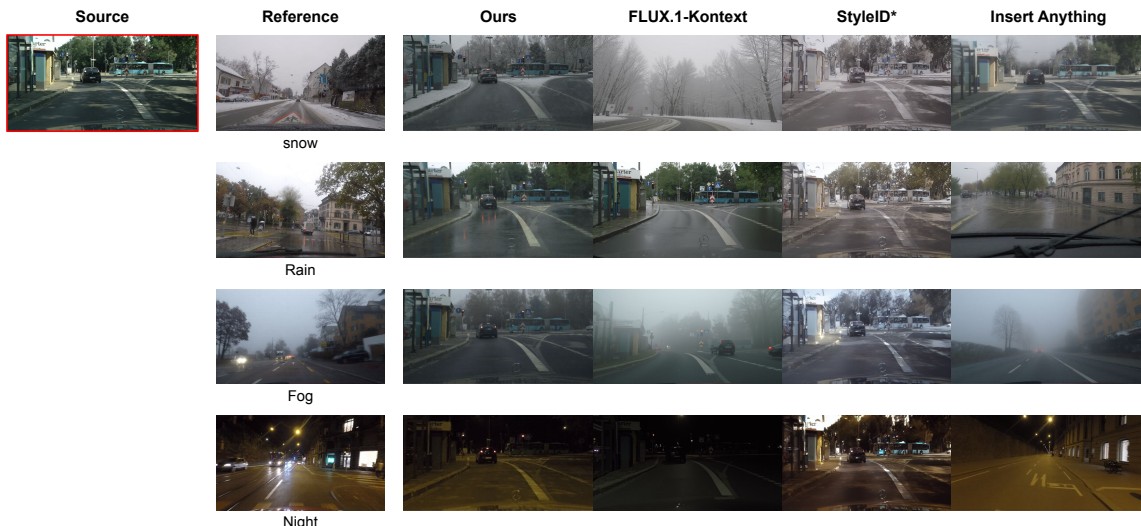

Figure 21: Qualitative comparison on the Cityscapes validation set.

**Reference**  **Source**  **Ours**

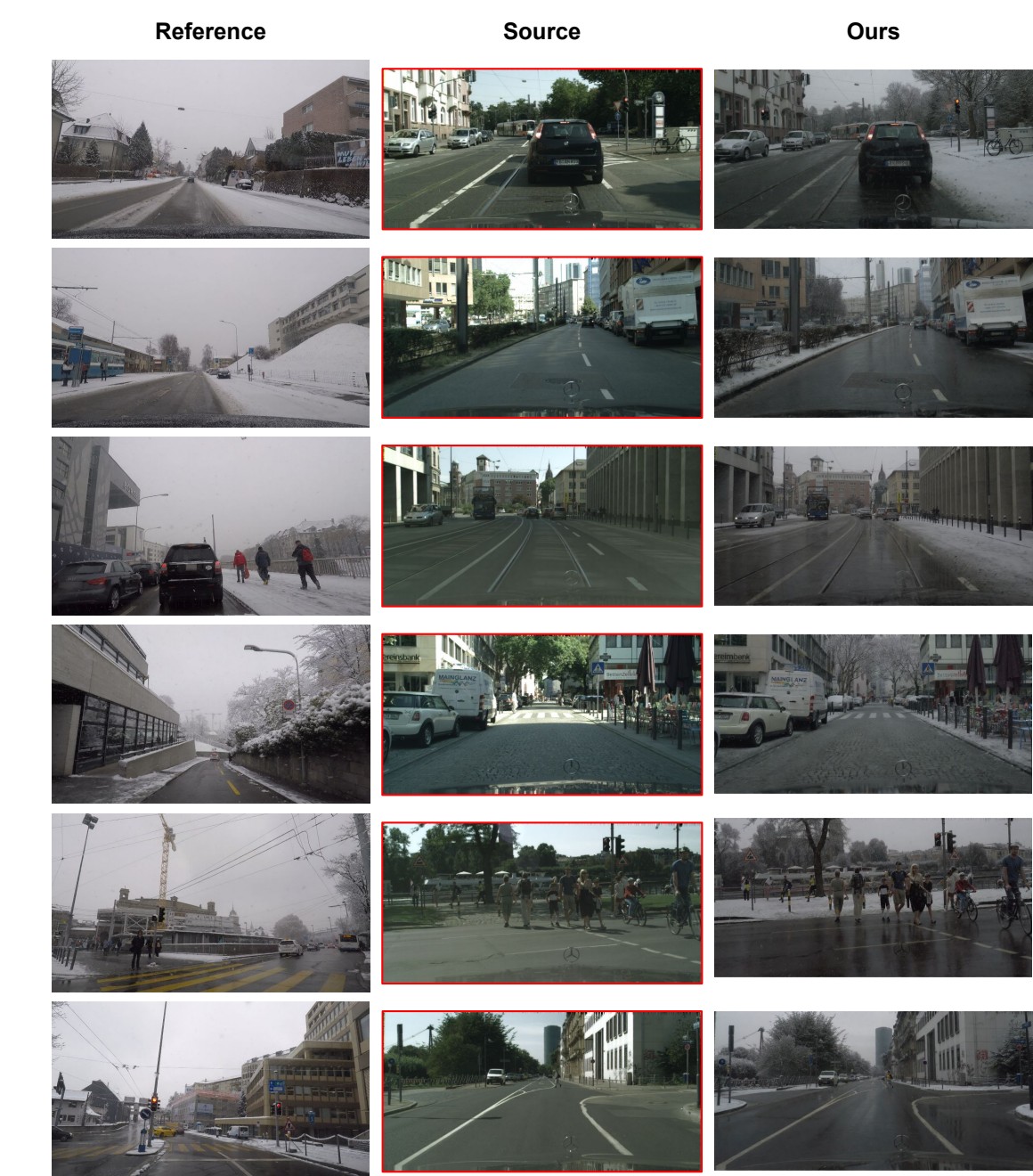

Figure 22: Sunny to snow weather translation on Cityscapes.

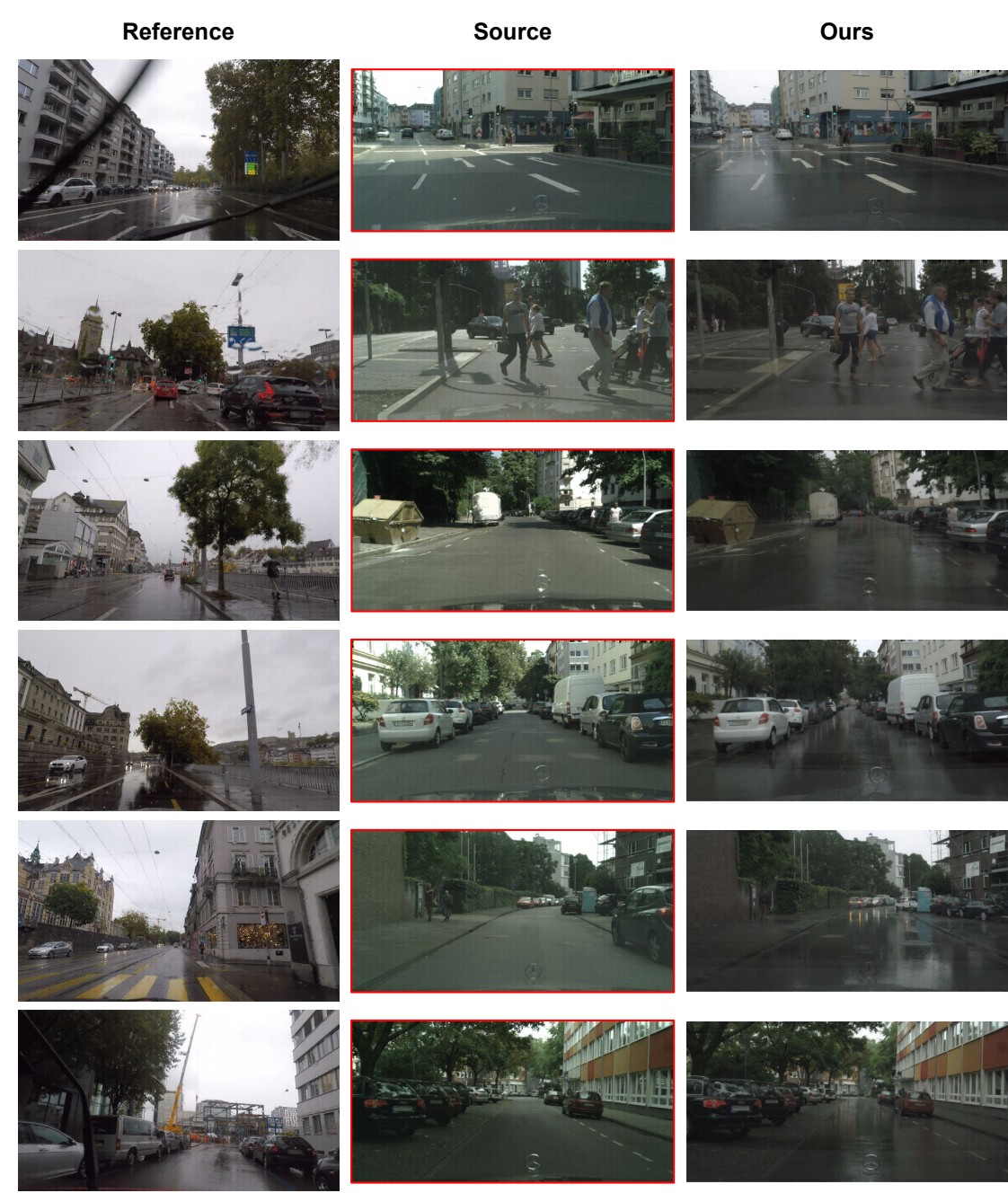

Figure 23: Sunny to rain weather translation on Cityscapes.

Reference         Source         Ours

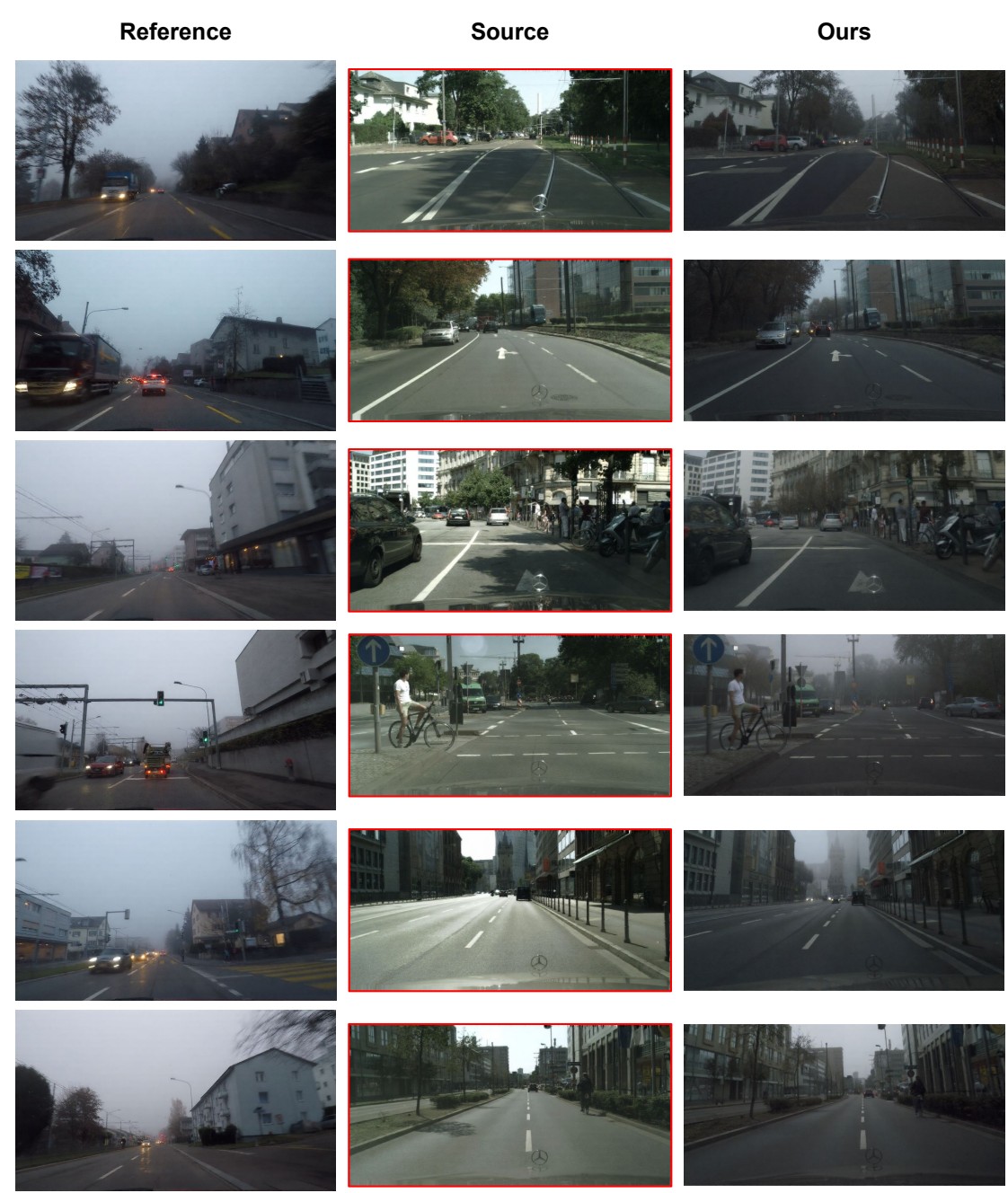

Figure 24: Sunny to fog weather translation on Cityscapes.

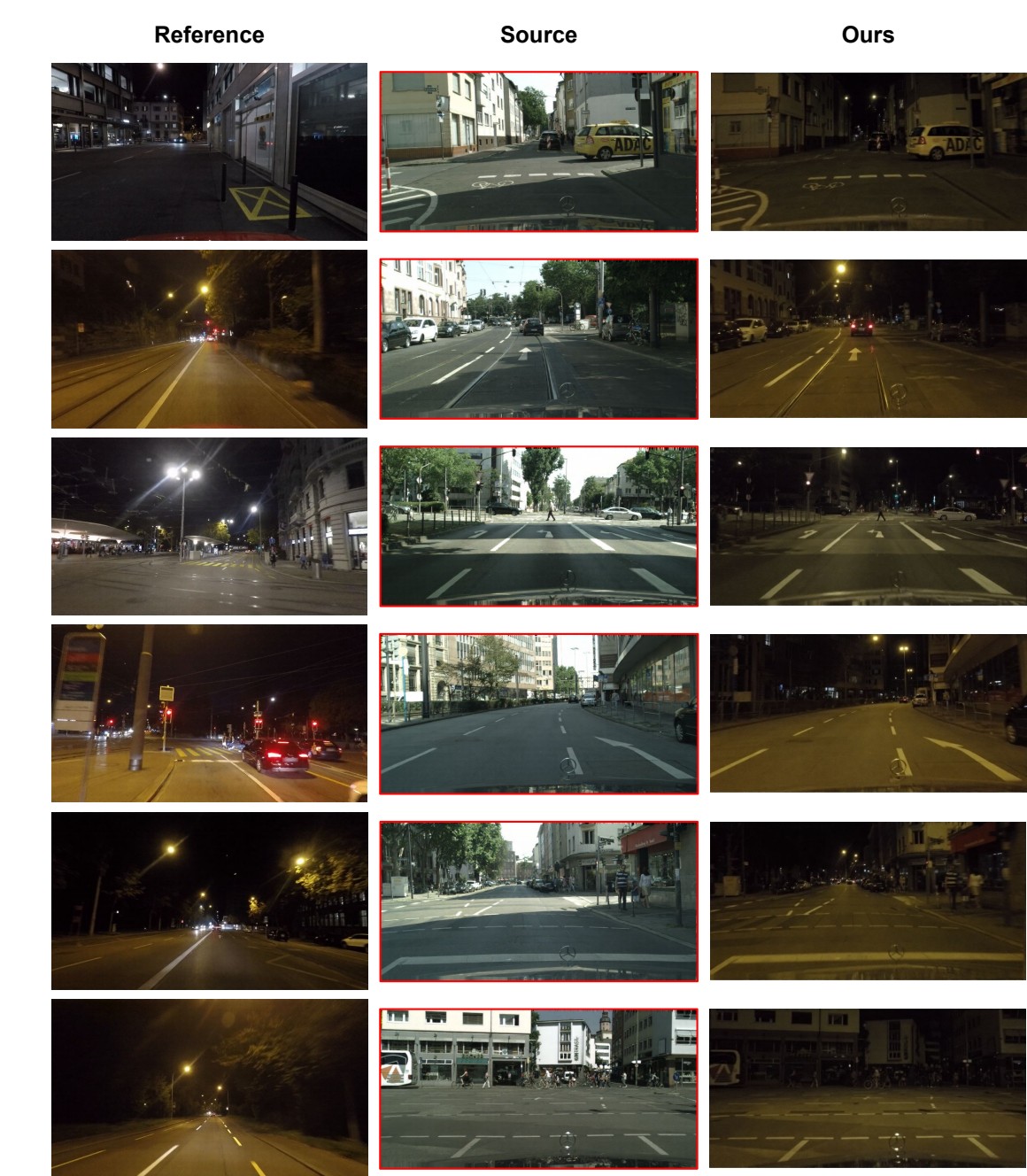

Figure 25: Sunny to night weather translation on Cityscapes.

