# OpenReview forum: "WeatherFLUX: Universal Weather Translation with Diffusion Models"
_ICLR.cc/2026/Conference — ICLR 2026 Conference Withdrawn Submission_

### Official Review · Reviewer_sbcp · 2025-10-30

**Soundness:** 3
**Presentation:** 3
**Contribution:** 2
**Rating:** 4
**Confidence:** 4

**Summary:**

This paper presents WeatherFLUX, a diffusion-based framework designed for universal weather translation in road scene images. The model enables bidirectional transformations between sunny conditions and other weather types, including snow, rain, fog, and night. WeatherFLUX addresses the challenge of dataset scarcity under diverse weather conditions, which is critical for autonomous driving research. To ensure structural consistency and semantic fidelity during translation, the authors introduce three key techniques, including image alignment, frequency-aware prompting, and semantic preservation loss. Extensive experiments demonstrate that WeatherFLUX produces high-quality, photorealistic translations while effectively preserving semantic content.

**Strengths:**

1) The paper makes a notable contribution by presenting the first universal diffusion-based model for realistic weather translation capable of functioning effectively with limited data. The proposed three designs in this work introduce a comprehensive solution to complex content preservation challenges in diffusion-based image-to-image translation.

2) Moreover, the authors conduct thorough experimental evaluations against several state-of-the-art baselines. The results show that WeatherFLUX consistently outperforms competing methods in visual realism and semantic preservation. This strong empirical performance underscores the technical soundness and novelty of the proposed approach.

**Weaknesses:**

1) Lacking the downstream evaluation to demonstrate the effectiveness of the synthesized data. Although the paper convincingly demonstrates improvements in image generation quality, it does not evaluate the practical effectiveness of the synthesized data for downstream perception tasks. Since one of the central motivations of this work is to alleviate data scarcity in adverse weather scenarios, it would be valuable to assess the utility of WeatherFLUX-generated images as the training data in tasks such as pedestrian detection, semantic segmentation, or 3D vehicle detection under different weather conditions. Such evaluations would provide stronger evidence of the method’s real-world applicability. Missing the evaluation on the downstream visual perception tasks would be one killing point to this paper.

2) Furthermore, it would be insightful to explore **which weather conditions yield the greatest benefit** when synthetic data are used for training, and to provide a deeper analysis of why synthetic datasets improve performance in specific challenging weather scenarios. These additional studies would enhance the paper’s impact and help clarify the broader applicability of WeatherFLUX.

**Questions:**

My main concern regarding this paper is that the authors did not perform the corresponding visual perception tasks on the downstream tasks  to really demonstrate the generated images could be used for promoting the perception performance. Only the evaluation regarding the image generation quality is not enough and heavily limit its real-world contribution.

---

### Official Review · Reviewer_5xF6 · 2025-10-31

**Soundness:** 2
**Presentation:** 3
**Contribution:** 3
**Rating:** 6
**Confidence:** 4

**Summary:**

The topic of this paper is about autonomous driving image generation under multiple weather conditions. The authors propose a diffusion-based framework WeatherFLUX by bidirection translation, which consists of image alignment, frequency-aware prompting, and semantic preservation loss. The proposed method focuses on preserving the semantic content of source image and the style of the reference image. WeatherFLUX has been evaluated on sereral datasets.

**Strengths:**

+ The paper is well-written and easy to follow.
+ The performance of the proposed method outperforms that of other methods by a margin.

**Weaknesses:**

- Lack of important comparison to related works. Some previous works (e.g., [a]) also utilize a diffusion-based framework to generate autonomous data. With a small amount of data, these works fine-tunes a pre-trained DiT network on multi-weather gerenation task. It is necessary to compare to these works. The difference among these works and the advantages of the WeatherFLUX can be highlighted.

[a] DriveDiTFit: Fine-tuning Diffusion Transformers for Autonomous Driving Data Generation, TOMM2025
- Evaluate the quality of the generated data. The authors mainly utilize the metric of image translation to evalute the quality of the generated data. To better show the effectiveness of the generated data, the authors can utilize the generated data to train a model on some paractical tasks (e.g., object detection). If the generated data is useful, the performance of trained model can be improved by using the mixture of generated and real data.
- The generalization ability of the proposed method.  The proposed WeatherFlux is based on FLUX.1-Fill. Can these proposed modules (image alignment, frequency-aware prompting, and semantic preservation loss) be effective on classical image translation tasks or utilizing different pre-trained open-source image generation models? It is important to demonstrate that these modules are tailored for mutiple weather autonomous driving image generation.

**Questions:**

My major concerns are included in the above weaknesses.

---

### Official Review · Reviewer_rEPC · 2025-10-31

**Soundness:** 2
**Presentation:** 2
**Contribution:** 2
**Rating:** 2
**Confidence:** 4

**Summary:**

In this paper, the authors propose a weather translation method that can directly translate road images from different weather condition pairwisely. Their method, WeatherFLUX, takes a stacked image composed by reference image, a source image, and a blank canvas as inputs, and a extra text prompt to formate the task. They propose three  techniques, i.e., image alignment, frequency-aware
prompting and a semantic preservation loss to  improve consistency and reduce these differences between source image and target image. The empirical study shows the superiority of their method over existing methods.

**Strengths:**

Here are the strengths of this paper:
1.The idea is easy and the paper is easy to follow.
2. The authors conduct empirical study and compare their method with other baselines.

**Weaknesses:**

This article exhibits critical issues in the following three key aspects:

1. The core components of the implementation method are notably missing. For instance, details such as how prompts are injected into the DIT+LoRA training process during training, and the specific structures adopted by each encoder/decoder module, are not clearly specified. This ambiguity makes it difficult to assess numerous technical details of the article. Furthermore, the authors have not provided any supporting code, rendering further evaluation of the method’s feasibility impossible. For example, the author mention that "For each weather condition the model is trained for 5000 iterations." Generally, it usually takes more iteration to train a diffusion model.

2. The frequency-aware prompting section proposes using Fourier transform to extract content and style information; however, high-frequency and low-frequency signals obtained via Fourier transform do not correspond to these two types of information, based on existing knowledge. More critically, the authors perform operations on features transformed by the encoder—this choice further obscures the exact correspondence between low-frequency/high-frequency signals and the intended information, making it harder to validate the technical rationale.

3.There is a contradiction in the method’s data dependency. While the authors emphasize reducing the reliance on paired data, the image alignment component of their approach still depends on the similarity of paired data to extract a sufficient quantity of aligned data. If the similarity between source and target data in the dataset is insufficient, their method fails to generate enough similar pairs. Additionally, the ACDC dataset used for training inherently contains a large amount of paired data, which makes it difficult to objectively judge the method’s actual performance in scenarios with limited paired data.

**Questions:**

Please check the weakness part and answer my major concern.

---

### Official Review · Reviewer_5bmA · 2025-11-02

**Soundness:** 2
**Presentation:** 2
**Contribution:** 1
**Rating:** 2
**Confidence:** 5

**Summary:**

This paper addresses the problem of translating images to new weather conditions. This is done by making the style and source images, and an empty canvas, available to a pre-trained diffusion model. The proposed approach relies on prompt learning using LoRA-based finetuning (as shown in Figure 2). Low-frequency feature components of the reference image, and thigh frequency counterpart of the source image are used to derive features which are later fused with the text prompt, to condition DiT for in-context learning. On the output side, the source image and the predicted image are compared by using the high-frequency components to maintain the semantic consistency between the input and output images. The method is developed on the ACDC dataset, and evaluation is primarily conducted using the FID metric.

**Strengths:**

* The paper addresses an interesting problem, the solution to which is likely to offer valuable contributions in robust perception in challenging scenarios.

* Paper is generally well written and easy to follow.

* The FID measure of the proposed method, semantic segmentation of Figure 6, as well as the qualitative results, are promising.

* The supplementary results are helpful, which also include some cityscape cases.

**Weaknesses:**

* This paper misses an ablation study, which makes it unclear which components are playing what role. It is unclear if the proposed prompting technique, the feature fusion, or the loss on the output side is playing the major role.

* Motivation for using the high-pass filter on the output side $f_H$ is not very convincing. It has been argued that the proposed loss ensures the semantic consistency between input and output images. First of all, the proposed loss does not measure any sort of semantics explicitly. Secondly, it is not clear if one wishes to maintain the exact semantics at all times. For example, tree branches in the winter are expected to have leaves during summer. Thirdly, no empirical evidence is provided to support the claim.

* The paper lacks exhaustive experiments and evaluations. It can be evaluated on more datasets and for many downstream tasks, including semantic segmentation. It would be interesting to see if the generated images could meaningfully serve as data augmentation.

* The reported FID claims of Tables 1 and 2 are not very convincing. This is mainly due to the lack of details regarding the FLUX.1 Kontext, which is also the backbone of the proposed method and the closest competitor.  Has FLUX.1 Kontext has been finetuned in the exact same setup of the proposed method? Could it be the case that all the performance gains come merely by fine-tuning FLUX.1 Kontext on the ACDC dataset. More importantly, within the context of the paper's experimental setting, more strong baselines could be established.

**Questions:**

* Please provide more details regarding the ablation study and attempts on stronger baselines.

* How does the $\mathcal{L}_{SP}$ maintain the semantics? Please provide the experimental evidence.

* Please report the failure cases and discuss limitations.

* What happens with the dynamic parts during the image alignment process?

---

### Note · Authors · 2025-12-03

I have read and agree with the venue's withdrawal policy on behalf of myself and my co-authors.